# Relationship between Perceived Teacher Support and Student Engagement in Physical Education: A Systematic Review

**Qi Guo** [1], **Shamsulariffin Samsudin** [1,*], **Xiaoming Yang** [1], **Jianxin Gao** [1], **Mohd Aswad Ramlan** [2], **Borhannudin Abdullah** [1] and **Noor Hamzani Farizan** [3]

[1]  Department of Sports Studies, Faculty of Educational Studies, Universiti Putra Malaysia, Serdang 43400, Malaysia; gs60639@student.upm.edu.my (Q.G.)
[2]  Department of Recreation and Ecotourism, Faculty of Forestry and Environment, Universiti Putra Malaysia, Serdang 43400, Malaysia
[3]  Defence Fitness Academy, National Defence University of Malaysia, Kem Sungai Besi, Kuala Lumpur 57000, Malaysia
*  Correspondence: shamariffin@upm.edu.my

**Abstract:** Student engagement is an essential indicator of educational quality and an instability behavior influenced by teachers. However, research on how teacher support influences this behavioral outcome in physical education has started late compared to academic settings. Hence, this systematic review aims to examine the empirical literature regarding the relationship between perceived teacher support and student engagement in physical education. This review complied with the PRISMA statement and ultimately found 11 eligible studies through the literature utilizing several databases, namely, Web of Science, SCOPUS, PsycINFO, ERIC, and SPORTDiscus. The results revealed a significant positive relationship between perceived teacher support and multiple dimensions of student engagement in physical education, especially in behavioral and emotional engagement. Mediating effects were found in five studies, and autonomous motivation and psychological need satisfaction were the crucial mediators. Teacher support plays a vital role in positive student–teacher interactions and students demonstrate enhanced engagement in physical education learning when teachers provide autonomy, competence, and emotional support. This study has great significance for developing instructional strategies to improve the sustainability of student engagement in physical education and also provides insight for researchers exploring options for optimizing motivational teaching styles to promote the development of sustainable teaching practices.

**Keywords:** teacher support; supportive teaching; student engagement; physical education

## 1. Introduction

Physical education (PE) is an important curriculum for developing an active and healthy lifestyle. Adolescents can improve their mental and physical health by engaging in regular exercise [1]. Although regular physical activities can effectively promote physical health, the participation rate of adolescents is usually lower than recommended by WHO guidelines [2]. However, lots of students have insufficient positive experiences in PE [3,4], commonly demonstrating a lack of interest in physical activity, a lack of motivation, and a lack of engagement [5,6]. Awareness among researchers and teachers of the status of student engagement in PE classes is critical in order to address these issues.

With the development of positive psychology, researchers have shifted from a focus on problematic psychology and behavior to an exploration of the positive and potentially constructive power of individuals, and the concept of student engagement has emerged [7]. A significant goal of PE lessons is to encourage students to engage in physical activity [8]. Student engagement is viewed as a critical educational and behavioral outcome in comprehending students' motivational processes in PE [9,10]. Engagement of students is a multi-faceted concept that can be broken down into three categories: behavioral, emotional,

and cognitive [11]. Behavior engagement involves how well students pay attention in class, ask questions, perform exercises consciously, and demonstrate positive responses. Emotional engagement involves a variety of affective responses in the classroom; for instance, interest, focus, enjoyment, happiness, sadness, and attitudes toward school. Cognitive engagement is an individual's investment in learning, the acquisition of new knowledge and abilities, and the use of self-regulation mechanisms [12]. In a subsequent study, Reeve and Tseng [13] added agentic engagement to explore the extent to which students are initiatively involved in the teacher's instruction.

In classroom-based instructional settings, teacher support has been shown to increase student engagement [14]. According to Jang et al. [15], there is a considerable link between student engagement, competence support, and autonomy support. In addition, Shih [16] also discovered that among Taiwanese eighth-graders, teacher autonomy support predicted intrinsic motivation as well as emotional and behavioral engagement. The role of student engagement in academic set is well recognized by academics. In a similar vein, student engagement is viewed as an essential motivating behavior in PE because it indicates goal-directed, sustained, and intense interactions with learning tasks [17], with Trowler [7] arguing that the importance of student engagement is "No longer questioned."

Teacher support is essential to the positive interaction between students and schools [18]. There are different definitions of the conceptual content of teacher support. In the social support model, teacher support is when teachers assist students in any context with informational, instrumental, emotional, and appraisal support [19]. In the past two decades, educational academics have adopted the self-determination theory (SDT) in large numbers [20]. SDT defines teacher support as the teacher's responsibility in activating and satisfying students' psychological needs and ultimately guiding them toward acquiring greater autonomy in their decision-making and improving their intrinsic motivation during the learning process [21]. In the basic psychological needs mini-theory (BPNT) of SDT, for humans to be highly motivated to develop and achieve optimal performance, one should satisfy the basic psychological needs of autonomy, competence, and relatedness [21–23].

Teacher support takes the form of autonomy, competence, and relatedness support, which correspond to students' basic psychological needs [24]. Autonomy support means that the teacher considers the problem from the student's perspective and reduces compulsive behavior toward the student, helping students feel they have the freedom and independence to make their own decisions and choices, allowing the expression of opposing viewpoints, and using inviting language to facilitate the internalization process [25,26]. Teacher competence support refers to establishing a clear structure in teaching and promoting students to make great efforts as much as possible to obtain the ideal learning results, increasing their propensity to feel competent in relevant learning tasks [15,27]. Teacher relatedness support involves teachers fostering intimate teacher–student connections in which students are cared for and shown unconditional affection [28]. Therefore, SDT provides a solid theoretical foundation for comprehending the positive consequences of teacher support. SDT is widely used as an essential theoretical framework in PE research because it reflects information on teaching behavior in PE and can be used to examine student engagement [29–31].

The aim of our systematic review was to examine the relationship between perceived teacher support and student engagement in the PE context. In traditional PE teaching, teachers have been accustomed to managing students' behavior in a controlled manner through monitoring, rewarding, and evaluating, so the study of supportive teaching styles can enlighten PE teachers in their teaching practices [32]. PE is usually more focused on teacher–student interaction and cooperation between students, and the role of the teacher is crucial to achieving the sustainability of teaching practices. Meanwhile, in many academic programs, student's abilities are frequently veiled, while in PE programs, they may be relatively overt. Accordingly, it is valuable to conduct such relevant research in the curricular setting of PE, and teacher support is an inescapable prerequisite for many of the key outcomes in educational psychology research. Increasing student engagement

improves academic performance, benefits students' future physical and mental health and social adjustment, decreases behavioral issues, and saves students from dropouts [33].

Despite the fact that the positive influence of teacher support on engagement has received much scholarly attention, the topic lacks a comprehensive review from the PE perspective. Therefore, this study utilized a systematic approach to conduct an extensive literature search and review of the empirical evidence for the relationship between perceived teacher support and student engagement within PE.

## 2. Materials and Methods

### 2.1. Eligibility Criteria

In this study, the Preferred Reporting Items for Systematic Reviews and Meta-Analyses (PRISMA) statement were employed as a guideline [34]. This study has been registered on the Platform of Registered Systematic Review and Meta-analysis Protocols (ID: IN-PLASY202250143). The below four inclusion criteria were used in this study to ensure that the related literature was obtained on the relationship between teacher support and student engagement in the PE context.

1. Articles published in English as well as in peer-reviewed journals;
2. Any cross-sectional, longitudinal, or experimental studies; reviews, theses and books were excluded;
3. No participant with a specific disability or other adverse physical or mental condition was included in the study;
4. Studies which investigated the relationship between perceived teacher support and student engagement in a PE context.

### 2.2. Search Strategy

We have conducted a comprehensive search for existing literature on the association between teacher support and student engagement in PE, published prior to 17 April 2022. A peer-reviewed literature search based on the abstract was performed independently by Guo and Yang across five databases, namely, Web of Science, SCOPUS, PsycINFO, ERIC, and SPORTDiscus. Keywords are referenced with relevant citations and terms from other researcher's systematic literature [35]. Every database was queried with the following keyword combinations: (AB = ("teacher support" OR "social support" OR "instructional support" OR "emotional support" OR "affective support" OR "autonomy support" OR "relatedness support" OR "teacher–student relationship *" OR "competence support" OR "structure" OR "supportive teaching" OR "relatedness" OR "competence" OR "autonomy") AND AB = ("student engagement" OR "student investment" OR "learning engagement" OR "student involvement" OR "cognitive engagement" OR "student participation" OR "behavioral engagement" OR "agentic engagement" OR "emotional engagement" OR "engagement" OR "social engagement") AND AB = ("physical education")).

### 2.3. Study Selection

First, articles from the literature search results were collected, and duplicates were removed using Zotero 6.0. Subsequently, Guo and Yang independently filtered the literature by the title and abstract, then screened it using their respective full texts. Those studies deemed not eligible were removed, and those that met the eligibility criteria were added accordingly. All reviewers agreed on the final inclusion of the studies in the systematic review, and consensus on any disagreements was reached through group discussions. Figure 1 describes a description of the procedures for inclusion according to the PRISMA statement.

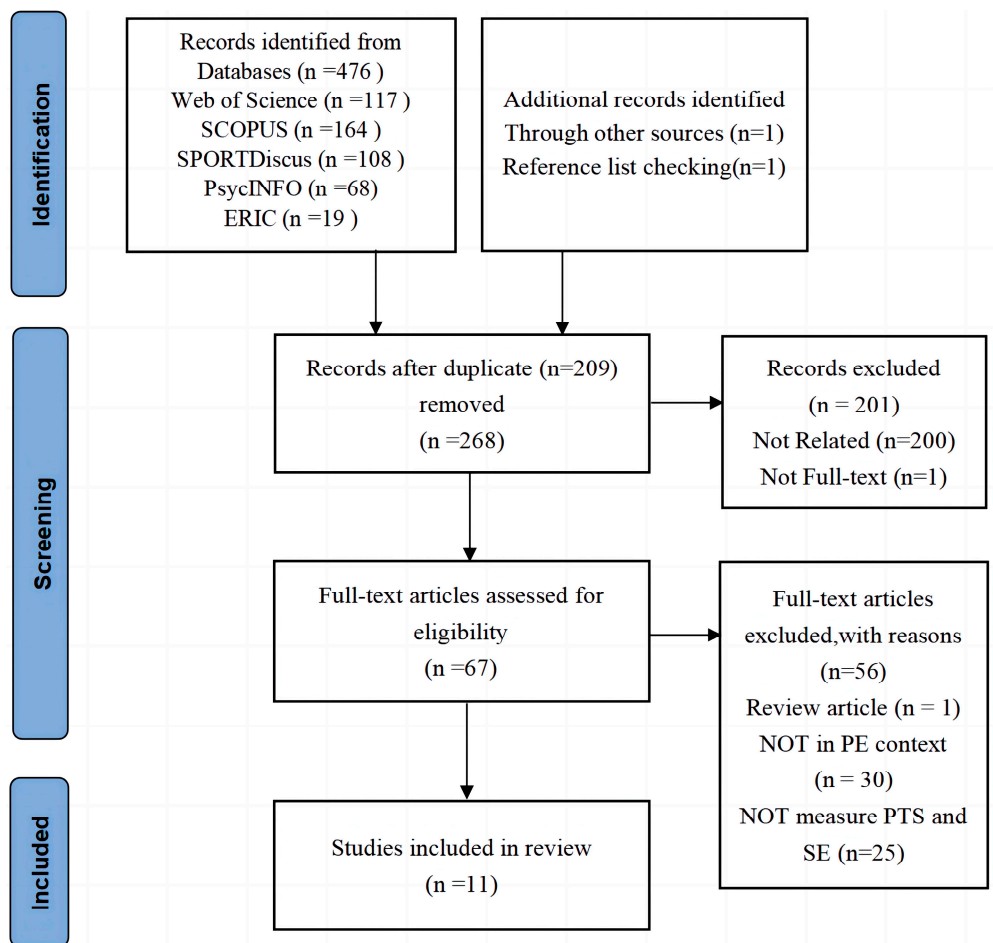

**Figure 1.** The PRISMA flow diagram.

*2.4. Quality Assessment*

In all eligible studies, risk of bias was assessed using the Quality Assessment Tool for Observational Cohort and Cross-Sectional Studies [36]. In previous systematic reviews, this tool was extensively utilized for assessing the quality of studies [37–40], which comprises the following 14 items: research question, eligibility criteria, study population, sample size, sufficient timeframe, exposure assessment, exposure of interest, repeated exposure assessment, exposure measures, blinding of outcome assessors, follow-up rate, confounders, statistical analyses, and outcome measures. The use of this tool allows us to specifically and validly assess the quality of the included literature. Overall quality ratings for this study were classified as good, fair, and poor. The two raters (Guo and Yang) individually rated each item and its overall quality, and disagreements were addressed by a third rater (Samsudin).

*2.5. Data Extraction*

In order to examine the methodologies and results of these studies, the full texts of the articles were read. We extracted and summarized the following information using a standardized template: (1) Author and Year; (2) Country; (3) Publications; (4) Study design; (5) Population and Grade level; (6) Sample size and Gender; (7) Age range and M ± SD; (8) Applied theory; (9) Measured dimensions; (10) Measures of perceived teacher support; (11) Measures of student engagement; (12) Main findings. The content of each included article is thoroughly analyzed and discussed separately by two review authors (Guo and Yang) and disagreements were worked out through discussion; if they disagreed, a third author (Samsudin) was consulted until they reached an agreement.

## 3. Results

*3.1. Study Characteristics*

We performed a search using predefined search terms. The database search identified 476 records: Web of Science (*n* = 117), SCOPUS (*n* = 164), SPORTDiscus (*n* = 108), PsycINFO (*n* = 68), ERIC (*n* = 19), and additional records were identified from a reference list (*n* = 1). After eliminating any duplicates, 268 records remained eligible for the screening process. Following a thorough review of the abstracts and titles, a total of 67 articles were eligible to be included. The full-text version of each article was assessed for eligibility. We have read the full texts of the manuscripts, and 56 were excluded for various reasons. For instance, 30 studies were not conducted in the context of PE, and 25 studies were without a measure of perceived teacher support or student engagement. In addition, one review article was excluded. Consequently, 11 studies were ultimately accepted for inclusion in this systematic review.

Among the articles evaluated, nine were good quality, and two were fair. Only two of the evaluated studies mentioned the method for calculating the sample size [41,42]. This raises questions about whether the sample is representative of the population. Eight of the total eleven articles utilized a cross-sectional design, which means that exposure and outcome were measured simultaneously; hence, there is no time for an effect to manifest, so the evidence on a causal relation between outcomes and exposures is weaker. Each of the studies, except two [43,44], applied an adequate and valid instrument to measure the variables they examined according to their research objectives. Table 1 summarizes the quality assessment results.

**Table 1.** Summary of quality assessment.

| Ref. | Criteria | | | | | | | | | | | | | | Rating |
|---|---|---|---|---|---|---|---|---|---|---|---|---|---|---|---|
| | 1 | 2 | 3 | 4 | 5 | 6 | 7 | 8 | 9 | 10 | 11 | 12 | 13 | 14 | |
| Otundo and Garn, 2019 [45] | Yes | Yes | NR | Yes | No | NA | No | NA | Yes | No | Yes | NA | NA | Yes | Good |
| De Meyer et al., 2016 [46] | Yes | Yes | NR | Yes | No | NA | No | NA | Yes | No | Yes | NA | NA | Yes | Good |
| Yoo, 2015 [47] | Yes | Yes | NR | Yes | No | NA | No | NA | Yes | No | Yes | NA | NA | Yes | Good |
| Gairns et al., 2015 [48] | Yes | Yes | NR | Yes | No | NA | No | NA | Yes | No | Yes | NA | NA | Yes | Good |
| González-Peño et al., 2021 [49] | Yes | Yes | NR | Yes | No | NA | No | NA | Yes | No | Yes | NA | NA | Yes | Good |
| Shen et al., 2012 [50] | Yes | Yes | NR | Yes | No | NA | No | NA | Yes | No | Yes | NA | NA | Yes | Good |
| Reeve et al., 2020 [41] | Yes | Yes | Yes | Yes | Yes | NA | Yes | NA | Yes | Yes | Yes | NA | NR | Yes | Good |
| Coterón et al., 2020 [51] | Yes | Yes | NR | Yes | No | NA | No | NA | Yes | No | Yes | NA | NA | Yes | Good |
| Leo et al., 2022 [44] | Yes | Yes | Yes | Yes | No | NA | No | NA | NR | No | Yes | NA | NA | No | Fair |
| Leo et al., 2020 [43] | Yes | Yes | NR | Yes | No | NA | No | NA | NR | No | Yes | NA | NA | Yes | Fair |
| Cheon et al., 2012 [42] | Yes | Yes | Yes | Yes | Yes | NA | Yes | NA | Yes | Yes | Yes | NA | NR | Yes | Good |

Note: NR, not reported; NA, not applicable.

The publication years of all included studies were within the interval of 2012 to 2022. These articles were primarily published in journals related to education and psychology. The highest percentage of articles published were found in the International Journal of Environmental Research and Public Health, with a total of two articles. Furthermore, the remaining publications appeared in a wide variety of periodicals, mostly from psychology journals such as Frontiers in Psychology and the International Journal of Educational Psychology. We also identified several publications related to PE, for example, in Physical Education and Sport Pedagogy, Journal of Teaching in Physical Education, etc.

The self-determination theory (SDT) was the foundational theory of nine studies because it addresses how the psychological aspects of students are influenced by the learning environment as well as the teacher and is more explanatory and supportive of such investigation [41–44,46–49,51]. The remaining examined research utilized either cognitive evaluation theory [47] or interest theory [45]. Shen et al. [50] employed the self-system model of motivational development (SSMMD) as the framework for their research. Publications and theories of the included studies are shown in Table 2.

**Table 2.** Publications and theories of included studies.

|  | No. of Article |
|---|---|
| Publications and Ranking |  |
| International Journal of Environmental Research and Public Health | 2 |
| International Journal of Educational Psychology Q3 | 1 |
| Psychology of Sport and Exercise Q1 | 1 |
| Perceptual and Motor Skills Q4 | 1 |
| Frontiers in Psychology Q1 | 1 |
| Journal of Teaching in Physical Education Q1 | 1 |
| International Journal of Behavioral Development Q3 | 1 |
| European Physical Education Review Q1 | 1 |
| Physical Education and Sport Pedagogy Q1 | 1 |
| Journal of Sport and Exercise Psychology Q2 | 1 |
| Theories |  |
| Self-determination theory (SDT) | 9 |
| Cognitive evaluation theory | 1 |
| Interest theory | 1 |
| Self-System Model of Motivational Development (SSMMD) | 1 |

The most frequently investigated dimension of perceived teacher support was autonomy support, followed by relatedness and competence engagement. The three articles examined the perception of teacher support based on three dimensions [43,44,50]. Regarding student engagement, behavioral engagement was studied by most studies. Five studies only focused on behavioral engagement [46–49,51]. Four studies only focused on the behavioral and emotional engagement [43–45,50]. Additionally, two studies focused on the less frequently used agentic engagement [41,42] and cognitive engagement [42]. The measured dimensions of the included studies are shown in Table 3.

**Table 3.** Measured dimensions of included studies.

| Ref. | Dimensions of PTS | | | Dimensions of SE | | | |
|---|---|---|---|---|---|---|---|
|  | **AS** | **CS** | **RS** | **BE** | **CE** | **EE** | **AE** |
| Otundo and Garn, 2019 [45] | ✓ | ✓ | ✓ | ✓ |  | ✓ |  |
| De Meyer et al., 2016 [46] | ✓ |  |  | ✓ |  |  |  |
| Yoo, 2015 [47] | ✓ |  |  | ✓ |  |  |  |
| Gairns et al., 2015 [48] |  |  | ✓ | ✓ |  |  |  |
| González-Peño et al., 2021 [49] | ✓ |  | ✓ | ✓ |  |  |  |
| Shen et al., 2012 [50] | ✓ |  | ✓ | ✓ |  | ✓ |  |
| Reeve et al., 2020 [41] | ✓ |  |  |  |  |  | ✓ |
| Coterón et al., 2020 [51] | ✓ |  |  | ✓ |  |  |  |
| Leo et al., 2022 [44] | ✓ | ✓ | ✓ | ✓ |  | ✓ |  |
| Leo et al., 2020 [43] | ✓ | ✓ | ✓ | ✓ |  | ✓ |  |
| Cheon et al., 2012 [42] | ✓ |  |  | ✓ | ✓ | ✓ | ✓ |

Note: PTS, perceived teacher support; AS, autonomy support; CS, competence support; RS, relatedness support; SE, Student engagement; BE, behavioral engagement; CE, cognitive engagement; EE, emotional engagement; AE, agentic engagement.

Survey questionnaires were utilized in all studies under review to measure different types of teacher support and student engagement. In most cases, items were adopted or adapted for measuring both variables using rating a 5- or 7-point Likert scale. Typically, scales developed by Standage et al. [52] and Learning Climate Questionnaire (LCQ) [53] were utilized for measuring perceived teacher support. For measures of student engagement in the included articles, The Engagement Questionnaire in Physical Education [50] and Engagement vs. Disaffection with Learning Scale (EVDLS) [54] are frequently employed in the included articles. Remarkably, in two articles based on the description of the perceived teacher support measure, the source of their citations could not be located and therefore it could not be confirmed whether the questionnaire they cited was validated in

the relevant population, which may affect the validity of the questionnaire [43,44]. The measurements of the included research are given in Table 4.

**Table 4.** Measurements of included studies.

| Ref. | Measure of PTS | Measure of SE |
|---|---|---|
| Otundo and Garn, 2019 [45] | Scales created by Standage et al. [52] | Engagement vs. Disaffection with Learning Scale (EVDLS) [54] |
| De Meyer et al., 2016 [46] | Teacher As Social Context Questionnaire (TASCQ) [29] | Scales created by Skinner et al. [55] |
| Yoo, 2015 [47] | Perceived Autonomy Support Scale for PE settings [56] | Engagement Questionnaire in PE [50] |
| Gairns et al., 2015 [48] | Scales created by Standage et al. [52] | Scales created by Ntoumanis [57] used for teacher's report |
| González-Peño et al., 2021 [49] | Scales created by Van den Berghe et al. [58] | Scales created by Reeve et al. used for observer [59] |
| Shen et al., 2012 [50] | Self-report relatedness scale [14] and Perceived locus of causality questionnaire [60] | Engagement vs. Disaffection with Learning Scale (EVDLS) for both used teacher's report and student's report [54] |
| Reeve et al., 2020 [41] | Learning Climate Questionnaire (LCQ) [53] | Agentic Engagement Scale [61] |
| Coterón et al., 2020 [51] | Scales created by Ruiz [62] | The Engagement Questionnaire in PE [50] |
| Leo et al., 2022 [44] | Teaching Interpersonal Style Questionnaire in Physical Education | Engagement vs. Disaffection with Learning Scale (EVDLS) [54] |
| Leo et al., 2020 [43] | Teaching Interpersonal Style Questionnaire in Physical Education | Engagement vs. Disaffection with Learning Scale (EVDLS) [54] |
| Cheon et al., 2012 [42] | Learning Climate Questionnaire (LCQ) [53] | Engagement vs. Disaffection with Learning Scale (EVDLS) [54], Scales created by Wolters [63], Scales created by Reeve and Tseng [13] |

Most studies were carried out in Spain (3) and South Korea (3). This is followed by the United States (2), Australia (1), Argentina (1), Belgium (1), and Columbia (1). González-Peño et al. [49] performed their research in two different countries: Spain and Argentina, respectively. There were eight cross-sectional studies, two experimental studies, and one longitudinal-experimental study. Cross-sectional studies were the most commonly used study design in reviewed studies. In all, 6698 samples in the age range of 10 to 22 years old were involved in the reviewed studies. The sample sizes ranged from 184 [50] to 2065 [44] participants. Most samples had both genders, but two samples only contained female participants [48,50]. In addition, one sample did not report the gender of student participants [49]. In terms of participants' education level, three studies focused on secondary school students, three on high school students, four studies included students at different educational levels, and one study did not report participants' educational levels [49]. Data extraction of the included study is summarized in Table 5.

**Table 5.** Data extraction of the included study.

| Author (Year) | Country | Study Design | Population/ Grade Level | Sample Size /Gender | Age (Range and M ± SD) | Main Findings |
|---|---|---|---|---|---|---|
| Otundo and Garn, 2019 [45] | USA | CS | 5 middle schools /6–8 grades | 388 F = 64% M = 36% | NR /12.40 ± 1.04 | Perceived AS, CS, and RS were associated with higher levels of students' BE and EE. |
| De Meyer et al., 2016 [46] | Belgium | RCT | 2 secondary schools /NR | 320 F = 214 M = 106 | 15–22 /17.28 ± 1.36 | Students in the AS group, relative to the controlling, reported more BE, and PNS mediated the effects. |
| Yoo, 2015 [47] | South Korea | CS | 8 middle schools /NR | 592 F = 288 M = 304 | 13–15 /14 ± 0.8 | Perceived AS was positively correlated with students' BE in PE; this relationship was mediated by AM. |
| Gairns et al., 2015 [48] | Australia | CS | 1 high school /7–10 grades | 374 F = 374 | 12–16 /13.36 ± 1.19 | Result showed a positive indirect relationship between RS and SE via positive teacher-focused RISE, RNS, and AM. |
| González-Peño et al., 2021 [49] | Spain and Argentina | CS | Schools in Buenos Aires and Madrid regions/NR | 709 /NR | 12–16 /19.16 ± 4.94 | RS could predict BE in a positive way and AS did not correlate with BE. |
| Shen et al., 2012 [50] | USA | CS | 3 high schools/9,10 grades | 184 F = 184 | 14–17 /15.1 ± NR | Teacher RS had significant effect on both students' BE and EE. Peers' RS moderated the effects of teacher RS on the student's BE. With AS as a covariate, teacher RS still significantly predicted the self-reported student EE. |
| Reeve et al., 2020 [41] | South Korea | RCT | 15 middle, 7 high schools/NR | 1422 F = 648 M = 773 | 13–18 /14.8 ± 1.6 | ASIP group students increased ANS and AE and decreased autonomy dissatisfaction and agentic disengagement; ANS and autonomy dissatisfaction mediated the effects of ASIP on AE and agentic disengagement. |
| Coterón et al., 2020 [51] | Colombia | CS | 27 private high schools /NR | 644 F = 53.1% M = 46.9% | 12–16 /15.16 ± 1.78 | Perceived AS was positively related to student's BE. |
| Leo et al., 2022 [44] | Spain | CS | 17 elementary and secondary schools /5th–11th grade | 2065 F = 1042 M = 1023 | 10–16 /11.96 ± 1.95 | Students of PE teachers with a high-low profile (high in perceived AS, CS, and RS and low in autonomy, competence, and relatedness thwarting) reported higher levels of BE and EE. |

**Table 5.** *Cont.*

| Author (Year) | Country | Study Design | Population/ Grade Level | Sample Size /Gender | Age (Range and M ± SD) | Main Findings |
|---|---|---|---|---|---|---|
| Leo et al., 2020 [43] | Spain | CS | 13 primary and secondary schools /5–11 grade | 1120 F = 561 M = 559 | 10–17 /11.7 ± 1.63 | Results showed a positive relation of AS, CS, and RS to students' BE and EE; this relationship was mediated by PNS and AM. |
| Cheon et al., 2012 [42] | South Korea | LG RCT | 18 middle schools, 3 high schools /NR | 1158 F = 550 M = 608 | NR | Teachers in the ASIP group displayed more AS; also, students perceived more AS and less controlling. ASIP intervention has a direct effect on students' BE, CE, EE, and AE; students' PNS partially mediated this effect. |

Note: CS, cross sectional study; RCT, randomized controlled study; LG, longitudinal study; F, female; M, male; NR, not reported; PE, physical education; PTS, perceived teacher support; AS, autonomy support; CS, competence support; RS, relatedness support; SE, Student engagement; BE, behavioral engagement; CE, cognitive engagement; EE, emotional engagement; AE, agentic engagement; ASIP, autonomy-supportive intervention program; RISE, relation-inferred self-efficacy; PNS, psychological need satisfaction; ANS, autonomy need satisfaction; RNS, relatedness need satisfaction; AM, autonomous motivation.

*3.2. Relationship between Perceived Teacher Support and Student Engagement in PE*

3.2.1. Direct Relationships

In the included studies, perceived teacher support was directly or indirectly associated with student engagement. The studies in which the relationship results are direct effects are as follows. Otundo and Garn [45] found a positive relationship between needs-supportive teaching and students' behavioral and emotional engagement in PE. Similarly, a recent large population-based cross-sectional study revealed that behavioral and emotional engagement is positively related to the three types of teacher support but negatively related to the three types of need-thwarting teaching styles, namely, controlling, cold, and chaotic teaching [44]. In the above two studies, perceived teacher support is measured by autonomy, competence, and relatedness support based on Deci's [64] core assumption of basic psychological needs. Cheon et al. [42] examined four types of student engagement in addition to teacher autonomy support. Different from other research, this study incorporates one additional agentic engagement of students. They found that an autonomy-supportive intervention program (ASIP) had a significant but modest direct effect on students' positive engagement.

Shen et al. [50] examined the distinctive contributions of teacher relatedness support; they found that students' perceived teacher relatedness had positive relationships with their emotional and behavioral engagement based on both self- and teacher-reports; relatedness towards peers also had an added effect on behavioral engagement. González-Peño et al. [49] found that relatedness support could predict behavioral engagement positively. However, he also reported that perceived autonomy support did not correlate with behavioral engagement, this finding is inconsistent with the other research included in the review. In contrast, Coterón et al. [51] and Yoo [47] found a significant positive association between behavioral engagement and perceived autonomy support, which means that the more autonomy support students perceived, the more behavioral engagement they demonstrated.

3.2.2. Indirect Relationships

Multiple studies have investigated the indirect relationships. Yoo [47] noted that autonomous motivation mediates, in part, the relationship between perceived autonomy support and behavioral engagement. The association between autonomous motivation and behavioral engagement was mediated by positive emotion. Another study involving only female participants demonstrated a positive indirect relationship between female students' perceptions of teacher relatedness support and their engagement via positive

teacher-focused relation-inferred self-efficacy (RISE), relatedness need satisfaction, and autonomous motivation [48]. Shen et al. [50] found that peer support for relatedness moderated the relationship between female students' relatedness toward teachers and behavioral engagement. In particular, students with poor perceptions of teacher relatedness exhibited significantly more behavioral engagement in PE if they felt acceptance and acknowledgment from their peers.

Leo et al. [43] showed that autonomous motivation and needs satisfaction mediate the relationship between perceived need-supportive teaching and engagement in physical exercise. Additionally, Cheon et al. [42] and De Meyer et al. [46] concluded that psychological need satisfaction is a mediator in the relationship between autonomy support instruction and student engagement. Meanwhile, there is also evidence that autonomy satisfaction mediated the effects of ASIP on agentic engagement [41].

These indirect relationships suggested a mediating role for psychological needs satisfaction and autonomous motivation in the relationship between teacher support and student engagement in PE lessons.

### 3.2.3. Experimental Studies

Three experimental investigations were found, and they were all RCTs [41,42,46]. In order to compare the efficacy of various approaches to education, De Meyer et al. [46] used an innovative video-based strategy. Three-hundred-and-twenty participants were randomly allocated to watch videos that were either controlled or autonomy-supportive teaching. The experiment lasted roughly 40 min and showed that students in the perceived autonomy condition reported higher behavioral engagement and less oppositional defiance than those in the controlled condition.

Additionally, two studies used the ASIP intervention program to assess its effectiveness on perceived teacher support and student engagement [41,42]. Cheon et al. [42] employed a longitudinal experimental design with follow-up surveys that included an experimental intervention. Observer evaluations revealed that after one semester of ASIP intervention, teachers in the experimental group demonstrated considerably more autonomy supportive instructional behaviors. According to student self-reports, teachers in the experimental group were perceived as more autonomy-supportive, and their students demonstrated higher behavioral, emotional, cognitive, and agentic engagement than students in the control group. Reeve et al. [41] observed that after an academic year ASIP intervention program, autonomy-supportive teaching directly improved both the levels of autonomy satisfaction and agentic engagement, and also reduced the levels of autonomy dissatisfaction and agentic disengagement.

### 3.3. Gender

Regarding gender differences among participants, Reeve et al. [41] found that male students scored higher on autonomy satisfaction and agentic engagement but lower on autonomy dissatisfaction and agentic disengagement than female students. Furthermore, Cheon et al. [42] discovered that male students scored better on perceived autonomy, perceived competence, and classroom engagement than female students. On the other hand, Leo et al. [44] found that females scored considerably better on perceived competence support, while males scored higher on emotional engagement. For male students, they were more prone to perceive the need-thwarting teaching of the PE teacher, whereas female students perceived more need-supportive styles. Leo et al. [43] found that compared with females, male students are more able to stave off need frustration with need-supportive teaching. In summary, gender differences in perceived teacher support and association with PE warrant further examination.

### 3.4. Grade Level

Only two research found disparities in the educational levels of participants. Reeve et al. [41] found that middle school students scored better than high school students

on perceived autonomy support, autonomy satisfaction, and agentic engagement. They also showed lower autonomy dissatisfaction and agentic disengagement. Cheon et al. [42] revealed that high school students scored higher on perceived autonomy than middle school students.

## 4. Discussion

The purpose of this research was to investigate for empirical evidence regarding the relation between perceived teacher support and student engagement in the PE setting. Following a comprehensive literature search, eleven papers establishing relationships between the two variables were gathered and examined.

More than half of the studies on this issue were undertaken in Spain and South Korea; the other studies are dispersed throughout many North and South American nations, and no relevant research was discovered from Africa. There is less research on gender and educational level differences in the study samples. In addition, all research recruited secondary or high school students. Engagement is frequently seen as a crucial aspect of educational reform goals, especially at the secondary level [12,65]. This suggests that there has been inadequate study conducted on the demographic of undergraduate students as well. "Students at the Heart of the System," a UK higher education white paper released in 2011, underlined student engagement as a crucial factor for the creation of learning communities in higher education [66].

The results of the analyzed studies indicated that all of the evaluated research papers employed questionnaires, bolstering the idea that quantitative research methods may be effective for examining perceived teacher support and student engagement. Two studies measured student engagement with teacher reports [48,50]. Compared to student self-reports, teachers' assessments tended to be more objective [54]. One research employed an indirect systematic observation [49], allowing for a deeper dive into how to improve supportive teaching to enhance student engagement.

In most studies, student engagement includes behavioral and emotional engagement; however, one study described student engagement across four dimensions, incorporating cognitive and agentic engagement in addition to the two types of engagement mentioned above [42]. Ten of the eleven included studies measured one of these dimensions, behavioral engagement. Since an essential observable indicator, behavioral engagement has particularly significant consequences for physical activity; it reflects some of the qualities that may encourage long term participation in physical exercise, such as effort, absorption, and persistence [67]. Nevertheless, future research needs to measure student engagement comprehensively to obtain a broader range of information and reveal more potential relationships, which would facilitate a more in-depth investigation of the relevant issues.

Our findings reveal a positive relationship between perceived teacher support and student engagement in PE, which is considerably greater than adverse or mixed outcomes. Most research grounded on SDT discovered a few partial connections among specific dimensions of each measure; for instance, between perceived autonomy support and behavioral, emotional engagement. Creating and supporting dynamic adaptation circumstances for the growth of student engagement may be achieved by fostering autonomy in a sports setting. Therefore, offering autonomy support efficiently is a crucial skill that PE teachers must learn [26]. According to our findings, ten studies validated the influence of perceived autonomy support on student engagement. As students have an intense sensation of the autonomy support provided by the PE teacher, they feel that classes are volitional and self-determined, leading them to participate gladly and voluntarily [45,47,48]. Consistent with the findings of a prior evaluation of classroom-based instructional contexts, it is generally believed that students are more likely to show significant and widespread improvement in learning outcomes when teachers provide higher levels of autonomy support [68].

In reaction to interactions with the social setting, one of the primary self-system processes that individuals acquire through time is a sense of relatedness [69]. Six studies examined the relation between perceived relatedness support and student engagement;

results indicated that a greater feeling of relatedness support was associated with an increase in emotional and behavioral engagement in PE [43–45,48–50]. Students who felt valued by their teachers would be more inclined to say their participation in the activities was amusing and enjoyable. Moreover, they would feel satisfied and confident in PE classes. In this scenario, they are more willing to expend efforts, focus intently, and stick to the learning task [50]. Teachers' pedagogical care inside the relational zone plays an essential impact in the motivating behaviors of students [70]. In conclusion, we propose that feeling connected and significant are not the byproducts of PE. Perceived relatedness support is crucial to the development of students. Teachers and educators should realize the importance of caring, address the needs of students, place students at the center of instruction, and attach significance to the classroom experience of students [71].

Three studies incorporated competence support as a distinct feature of perceived teacher support, allowing for a comprehensive examination of its influence on student engagement from the standpoint of SDT [43–45]. For example, Otundo and Garn [45] stated that students with more competence needs-supportive instruction showed more significant levels of personal interest, behavioral engagement, and emotional engagement. Obviously, there is limited research that integrates the dimension of teacher competence support. A possible reason for this is that competence support may not be expressed in the same manner in different contexts, and measuring competence consistently under different domains may be challenging.

Two studies expanded prior research by evaluating a motivational model that incorporated both the negative (i.e., need-thwarting teaching style) and positive (i.e., need-supportive teaching style) aspects of motivation [43,44]. In contrast to need-supportive teachers, need-thwarting teachers tend to apply strict disciplinary requirements and criticize students who do not perform to their expectations, eliciting feelings of guilt [72]. Leo et al. [43] concluded that students who saw their autonomy, competence, and relatedness demands met during PE classes reported greater levels of participation and greater willingness to engage in physical activity in their daily lives. Conversely, perceived need-thwarting teaching techniques are anticipated to diminish motivation in PE programs. Similarly, a recent study found that students who demonstrated significant behavioral and emotional engagement in PE classes were those who reported higher autonomy, competence, and relatedness support, along with lower autonomy, competence, and relatedness thwarting teaching [44]. Consistent with earlier research, the above conclusions imply that it is most beneficial for students when PE teachers support students' needs while avoiding practices that undermine their needs [73]. By comparing the opposites of supportive teaching, these findings highlight the importance of teachers' support for student engagement in PE classes. Given that the outcome variables in these two studies included only positive consequences, however, this may have resulted in an inability to fully assess the potential negative influences of the teaching styles on students. Future researchers should consider the effects of both needs-supportive and need-thwarting teaching styles on positive and negative outcomes for students in order to better understand the benefits and disadvantages of different teaching styles.

An autonomy-supporting intervention program (ASIP) designed and implemented by Cheon et al. [42] assists PE teachers in becoming more autonomy supportive during instruction. Initially, experienced educators (or instructors and coaches) were asked to engage in a training intervention based on the autonomy support principles of SDT. Teachers were then advised to adopt autonomy-supportive instruction over the course of many weeks or months. Thirdly, researchers evaluated the course-specific (or sport-specific) results experienced by their students or athletes, and the result showed that greater behavioral, emotional, cognitive, and agentic engagement was seen among students in the intervention group teachers' classes. Reeve et al. [41] also conducted ASIP in the study. After teachers, through ASIP, learned how to deliver better autonomy support in the classes, their students reacted by increasing agentic engagement while also decreasing agentic disengagement. Students of ASIP-participating teachers exhibited a variety of course-related advantages,

but it is still being determined how long-lasting the training-induced advances in teachers' capacity to be more autonomy-supportive can be. Therefore, future research should evaluate its long-term implications on student engagement, and PE teachers might benefit from being aware of their activity's influence on their students' conduct. Meanwhile, it is necessary to design and develop different types of intervention programs related to teacher support; thought-provoking evidence in this regard is required.

However, there are still contentious finding that contradict those reported by other studies that highlight the advantages of autonomy support. According to González-Peño et al. [49], no correlation was found between teachers' autonomy support and students' behavioral engagement. The observation instrument suggested by Van den Berghe et al. [58] includes certain autonomy dimension questions that may be comparable to the qualities of a prospective teacher. Thus, participants may view autonomy as a chaotic classroom in which the teacher has little control over what occurs, as is typical with an anticipating teacher. This uncertainty about these two aspects may explain why perceived teacher autonomy support and student engagement are unrelated.

Moreover, the results of this study showed that autonomous motivation and psychological needs satisfaction play a mediating role in the relationship between perceived teacher support and student engagement. In line with previous research, which revealed that an autonomy-supportive setting is important for class engagement through generating autonomous motivation, it is also generally advantageous [74]. According to SDT and earlier studies, psychological need satisfaction positively predicts engagement and intention in PE lesson [75–77]. Partial mediation by autonomous motivation and psychological needs satisfaction also reveals that additional factors may mediate the relationship between teacher support and student engagement in a PE environment, which could encourage future studies to explore other possible mediators.

Finally, our systematic research demonstrates that a supportive environment created by PE teachers is beneficial to promoting student engagement. Developing specific teacher training programs for the PE settings is valuable in educational practice. Meanwhile, several reviewed studies suggest some effective teaching strategies for PE teachers. Autonomy support strategies involve teachers being able to detect students' emotions during instruction, discussing possible outcomes and goals, listening to students' perspectives, vitalizing inner motivational resources, communicating in an inviting tone, providing explanatory rationales, and encouraging self-initiation by offering a variety of physical activity options [78,79]. Individualizing activities where feasible, delivering relevant encouragement during student practice and performance, providing feedback on their request, and taking into consideration a variety of skill levels are examples of competence-supportive tactics [45]. To increase support for relatedness, PE teachers should create a pleasant and motivating atmosphere during instruction, provide encouragement, respect students' individuality, and show amiability [43].

In summary, our systematic review found that teachers' support for students' basic psychological needs regarding autonomy, competence, and relatedness was positively related to student engagement. A comprehensive, evidence-based understanding of this relationship in the context of PE has been strengthened by the analysis of multiple findings. Teachers constructing a supportive PE environment facilitate the creation of positive teacher–student relationships and better understand and meet students' basic psychological needs, creating the conditions for motivational adaptation for the sustainable development of student engagement in PE programs. Our study highlights that teachers who provide a motivational teaching style (i.e., supportive teaching) are more likely to promote sustainable PE teaching practices. This study provides a reference for designing effective PE interventions to enhance students' sustainability of engagement and benefit their mental health development.

## 5. Conclusions

This review analyzed 11 systematically collected studies that provided empirical evidence for a relationship between perceived teacher support and student engagement in the PE context. In our review, Spain and South Korea accounted for more than half of the studies. Future research on this topic in other countries is needed to facilitate the further development of measurement instruments applicable to different cultural contexts. This will help fully understand the validity of the relationship hypothesis of these variables in the context of global differences in PE teaching. Most research samples are secondary school and high school students, and it is essential to conduct additional studies on the higher educated populations to confirm our findings.

The most frequently studied aspects of student engagement are behavioral engagement and emotional engagement, but dimensions such as agentic and cognitive engagement still need to be thoroughly explored. Present evidence indicates that teacher autonomy support, emotional support, and competence support are beneficial educational resources in the PE environment, which have a significant positive relationship with student engagement in PE. In other words, PE teachers can enhance students' engagement in PE lessons by encouraging decision-making, building positive teacher–student relationships, and promoting their ability to feel effective in their physical skills. However, many of these studies were cross-sectional, which prevents us from explaining the causal relationship between perceived teacher support and student engagement. Without sufficient experimental data, reliable conclusions cannot be derived.

Therefore, relevant PE educators and researchers should realize the importance of supportive teaching and focus on supporting students to meet their psychological needs and to promote student engagement. Further in-depth examination of the relationship between teacher support and student engagement in the PE context is imperative to improving students' learning achievements and enhancing the teaching ability of PE teachers. Lastly, in order to encourage students to stay engaged in physical activities, it is necessary to strengthen the design and implementation of long-term supportive teaching training programs and strategies.

## 6. Limitations

We restricted our search to articles written in English, so this review did not include articles written in other languages. In addition, another limitation of this study is the insufficient number of RCTs, most of which have a cross-sectional design limiting causal inference. Future research should be conducted with more longitudinal and experimental designs, which help to reveal causal relationships and enrich the scarce empirical data. In addition, most of the research assessed teacher support and student engagement based on student views; students' unconscious biases may affect the results. Therefore, future studies should circumvent this issue by incorporating ratings from external observers or using physiological sensors such as wrist-worn activity trackers to monitor students' physical reactions during class; this would enable researchers to obtain more reliable results regarding PE classes.

**Author Contributions:** Conceptualization, Q.G. and S.S.; methodology, Q.G., S.S., M.A.R. and B.A.; validation, S.S., M.A.R. and B.A.; formal analysis, Q.G.; investigation, Q.G. and X.Y.; resources, Q.G. and J.G.; data curation, Q.G. and X.Y.; writing—original draft preparation, Q.G.; writing—review and editing, Q.G., S.S., M.A.R. and N.H.F.; visualization, Q.G., X.Y. and J.G.; supervision, S.S and M.A.R. All authors have read and agreed to the published version of the manuscript.

**Funding:** This research received no external funding.

**Institutional Review Board Statement:** Not applicable.

**Informed Consent Statement:** Not applicable.

**Data Availability Statement:** Not applicable.

**Acknowledgments:** The authors would like to express their sincere gratitude to the reviewers and editors for their valuable suggestions and to Chengliang Wang for his careful guidance.

**Conflicts of Interest:** The authors declare no conflict of interest.

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
