# Peer review of "Relationship between Perceived Teacher Support and Student Engagement in Physical Education: A Systematic Review"

_sustainability, doi:10.3390/su15076039_

Round 1

Reviewer 1 Report

The title of the article is apt and clear. The abstract does not have the required structure (structured abstract) it needs to be revised according to the journal's criteria.
The introduction to the issue is sufficient, based on up-to-date and relevant literature sources.
The thesis lacks the aim of the processed review study, it is briefly stated at the end of the introductory part.
The methodology is described correctly but it is not clear from the data presented whether the reviewers worked independently in the search for articles, independently in the selection of articles. Other data in the Material and Methods section are described correctly and clearly according to the selected methodology.
The Results and Discussion section lists the most important findings from the studies presented. I think the article would be more interesting for the reader if these sections were divided into subsections according to the content.
The findings of the review paper are briefly summarized in the conclusion.  I would recommend adding the authors' own perspective and recommendations for practice (conducting studies in countries where they are absent, measurement tools, etc.).

Bibliography is not listed as recommended.

Author Response

Dear reviewer

Reviewer 2 Report

With academic papers, for me there are two important sides: technical soundness that allows readers to become convinced about the quality of the authors' work; and the message. 

From technical side, your work (from my point of view - other reviewers may have different opinions) was perfect. From the message side, I also find the paper good, but I would encourage you to set higher goals next time as it is obvious that you are technically able to reach them. 

Author Response

Dear Reviewer

Reviewer 3 Report

The manuscript addresses the relevant question of options for optimizing motivational teaching styles in physical education. Knowledge of the relevant state of research and the relevant literature is shown at an acceptable level.

The authors clearly adhere to the methodological approach, each variable is well substantiated, the number of tables and explanations allow the reader to understand the message of the authors.

At the same time, authors are encouraged to consider eliminating the following shortcomings:

- some paragraphs look too long, making it difficult for the reader to understand them

- keyword combinations chosen by the authors could be more substantiated

- “The purpose of this research was to look for empirical evidence regarding the relation between perceived teacher support and student engagement in the PE setting” – such a purpose lacks scientific novelty; the tasks of any review are obviously beyond the scope of the verb “to look”. Perhaps that is why in the Discussion section, the authors again (as in the previous Results section) declare the results more (but this time they operate with qualitative variables rather than quantitative ones how it was done in the Results section), instead of discussing and critically analyzing the selected publications. Stringency of the argumentation and the discussion could be reinforced.

Author Response

Dear Reviewer

Reviewer 4 Report

This article is well written and well structured, and presents a very interesting review that focus on an increasingly important issue, how can we teachers manage to captivate our students so that they dedicate themselves to their studies? In this specific study it is applied in the context of PE, but it could be applied in any other academic context.

The research carried out in this article seems to me to be in line with expectations, the references used are adequate and I believe that the objective of this study brings added value to research to this topic in particular.

The information in the tables was quite interesting and easy to understand, but regarding table 4, the measure and the respective description should be there instead of "just" the reference to the article.

In the discussion section I really liked the detail given to the different articles that were analyzed, and all the information present there is very well supported.

The conclusion could be more detailed in terms of metrics, but other than that, the information present here seems to be according to the expected.

I did not find any major errors, so after some minor corrections, it is an very interesting article.

Errors/questions/comments:

Line 175: "and two was fair" - maybe "were"?

Line 364: "across four dimensions" - The name of the dimensions should be indicated

Author Response

Dear Reviewer
